# Changes in precipitation and atmospheric N deposition affect the correlation between N, P and K but not the coupling of water-element in *Haloxylon ammodendron*

Zixun Chen[1], Xuejun Liu[1,2], Xiaoqing Cui[1,2], Yaowen Han[1], Guoan Wang[1,3]*

1 Beijing Key Laboratory of Farmland Soil Pollution Prevention and Remediation, Department of Environmental Sciences and Engineering, College of Resources and Environmental Sciences, China Agricultural University, Beijing, China, 2 Xinjiang Institute of Ecology and Geography, Chinese Academy of Sciences, Urumqi, China, 3 Key Lab of Plant–Soil Interaction, College of Resources and Environmental Sciences, China Agricultural University, Beijing, China

* gawang@cau.edu.cn

**Data Availability Statement:** All relevant data are within the manuscript and its Supporting information files.

## Abstract

Global changes in precipitation and atmospheric N deposition affect the geochemical cycle of the element and its hydrological cycle in the ecosystem. It may also affect the relationship between plant water use efficiency (WUE) and nutrients, as well as the relationship between plant nutrients. Desert ecosystems are vulnerable to global changes. *Haloxylon ammodendron* is the dominant species in the Asian desert. Revealing the variations in these relationships in *H. ammodendron* with precipitation and N deposition will enhance our understanding of the responses of plants to global change in terms of trade-off strategies of nutrient absorption, water and element geochemical cycles in desert ecosystems. Thus, we conducted field experiments with different amounts of water and N. This study showed that WUE of *H. ammodendron* was not correlated with nitrogen content (N), phosphorus content (P), and potassium content (K) when water and N supply were varied (p > 0.05 for WUE vs. N, P, and K), suggesting lack of coupling between water use and nutrient economics. This result was associated with the lack of correlation between plant nutrients and gas exchang in *H. ammodendron*. However, water addition, N addition and the interaction between both of them all played a role in the correlation between plant N, P and K owing to their different responses to water and N supplies. This indicates that global changes in precipitation and N deposition will affect N, P and K geochemical cycles in the Asian deserts dominated by *H. ammodendron*, and drive changes in the relationships between plant nutrients, resulting in changes in the trade-off strategy of plant absorption of N, P, and K.

## Introduction

Atmospheric N deposition has continued to rise since the Industrial Revolution [1–3], and the pattern of global rainfall has also changed [4, 5]. These changes significantly affected plant

**Funding:** This research was supported by the Chinese National Basic Research Program (No. 2014CB954202) and a grant from the National Natural Science Foundation of China (No. 41772171). The funders had no role in study design, data collection and analysis, decision to publish, or preparation of the manuscript.

**Competing interests:** The authors have declared that no competing interests exist.

resource utilization, hydrological cycles, and geochemical cycles of elements. Plant water economy is closely associated with water geochemical cycle, and nutrition economy is closely related to element geochemical cycle. To resist environmental changes, the adjustment of nitrogen, phosphorus, and potassium economies and water economy is an important strategy for resource utilization in plants. Nitrogen content (N), phosphorus content (P), and potassium content (K) in plants can serve as surrogates for the nitrogen, phosphorus, and potassium economies [6]. Many studies have demonstrated that plants N, P, and K are closely related with each other [7–11]. Water use efficiency (WUE) is the main indicator of plant water economy, and it is associated with gas exchange including photosynthetic rate ($A$), transpiration rate ($E$) and stomatal conductance ($g_s$). Plant N is positively correlated with $A$ because it is the main constituent of Rubisco (ribulose-1,5-bisphosphate carboxylase oxygenase) and chlorophyll in plants [12–14]. Phosphorus is required for many compounds involved in photosynthesis [15]; therefore, plant P is positively correlated with some parameters related to photosynthesis, such as maximal Rubisco carboxylation rate and maximum electron transport rate [16–18]. Potassium plays a crucial role in adjusting stomata, osmotic pressure and enzyme activity [19–21], and thus strongly controls the changes in $g_s$ and $E$. Therefore, WUE is expected to be related to plant N, P and K, and water use should be linked to plant N, P, and K economy [6]. Several studies have reported tight coupling between WUE and plant N, P and K [6, 22–27]. However, other investigations did not observe these couplings [23, 26, 28, 29]. The relationship between plant WUE and nutrients and the relationship between plant N, P and K reflect the coupling between water cycle and the geochemical cycles of these elements in ecosystems along with the resource-use strategies adopted by plants [30, 31].

Changes in rainfall and N deposition may also affect the relationship between plant water use and nutrient use, and the relationship between different nutrient elements. Two meta-analyses have found that global change, including change in N deposition and precipitation, have resulted in variations in plant N/P ratio and N/K ratio [32, 33], indicating that plant element coupling has been affected by changes in rainfall and N deposition. Under increasing precipitation, plants will take up more nutrients [22], whereas WUE will decrease [34]. Increasing N deposition promotes photosynthesis [12–14], causing plant WUE to increase. Despite this, due to the dilution effect of biomass, plant N, P, and K may not increase. Therefore, with N deposition and precipitation changes, the direction of changes in WUE and plant N, P, and K may be different, which will lead to changes in the correlation between WUE and plant N, P, and K. However, at present, only a few studies have investigated the influence of N deposition and precipitation changes on the relationship between WUE and plant N, P, and K.

Due to extreme drought and barrenness, desert ecosystems are vulnerable to environmental changes [35, 36]. Rainfall changes and elevated atmospheric N deposition are two important factors influencing the availabilities of water and N in deserts [37]. *Haloxylon ammodendron* is a dominant species in desert regions, particularly in Asian deserts. It plays an important role in the stabilization of sand dunes, the survival and development of understory plants, and the structure and function of desert ecosystems [38–40]. Given the universal plant WUE-nutrient coupling and the mutual relationships between plant N, P, and K [6.7.27], and the possible impact of changes in precipitation and N deposition on these couplings [32, 33], we hypothesized that: (1) plant WUE-nutrient coupling and couplings among plant N, P, and K should occur in *H. ammodendron*; (2) the correlation between plant nutrients should vary with changes in rainfall and atmospheric N deposition for *H. ammodendron*; and (3) changes in rainfall and atmospheric N deposition also have an effect on these correlations between WUE and N, P, and K for *H. ammodendron*. To test our hypotheses, we conducted a field experiment by varying the supply of water and nitrogen in the southern Gurbantunggut Desert in Xinjiang Uygur Autonomous Region, China. We measured WUE, N, P and K of the assimilating

branches of *H. ammodendron*. We hope this study can enhance the understanding of the resource-use strategies adopted by plants growing in desert ecosystems and the responses of water and element geochemical cycles to changes in N deposition and precipitation in desert ecosystems.

## Materials and methods

### Study site

This study was conducted at the Fukang Station of Desert Ecology, Chinese Academy of Sciences, on the southern edge of the Gurbantunggut Desert (44˚26′ N, 87˚54′ E) in northwestern China. The altitude of our study site is 436.8 m above average sea level (a.s.l.). It has a typical continental arid, temperate climate with a hot summer and cold winter. The mean annual temperature is 7.1 ˚C, and the mean annual precipitation is 215.6 mm, with potential evaporation of about 2000 mm. The soil type is gray desert soil (Chinese classification) with aeolian sands on the surface (0–100 cm). The percentages of clay ($<$ 0.005 mm), silt (0.005–0.063 mm), fine sand (0.063–0.25 mm), and medium sand (0.25–0.5 mm) ranged from 1.63% to 1.76%, 13.79% to 14.15%, 55.91% to 56.21% and 20.65% to 23.23%, respectively [41]. The soil is highly alkaline (pH = 9.55 ± 0.14) with low fertility. The vegetation is dominated by *Haloxylon ammodendron* and *Haloxylon persicum* with approximately 30% coverage. Herbs include ephemerals, annuals, and small perennials, covering approximately 40% [42]. Although the coverage of the two *Haloxylon* species is slightly lower than that of herbs, the biomass of the former is much larger than that of the latter. This is because *Haloxylon* plants are small trees with an average height of 1.5 m, whereas the latter are very low herbaceous plants. Biological soil crusts are distributed widely in the soil between herbs and *Haloxylon*, with approximately 40% coverage [43].

### Experimental design

A field experiment with a completely randomized factorial combination of water and nitrogen has been conducted since 2014. Two water addition levels (0, 60 mm·yr$^{-1}$; W0, W1) were chosen based on the fact that precipitation was predicted to increase by 30% in northern China in the next 30 years [44]. Three levels of N addition (0, 30, 60 kg N·ha$^{-1}$·yr$^{-1}$; N0, N1 and N2) were chosen based on the fact that N deposition has reached 35.4 kg N·ha$^{-1}$·yr$^{-1}$ in a nearby city, Urumqi [40] and is expected to double by 2050 relative to the early 1990s [45]. Therefore, six treatments (W0N0, W0N1, W0N2, W1N0, W1N1, and W1N2) were used in this experiment. Four replicates of each treatment were set, resulting in a total of 24 plots. The plots were distributed in a lowland area between two dunes. Each plot was 1.5 m×1.5 m with a *H. ammodendron* enclosed in the center. The average height and coverage of an individual *H. ammodendron* were 1.5 m and 1.9 m$^2$, respectively; they did not vary significantly across the plots. To simulate natural rainfall and N deposition, water and N additions were applied in equal amounts, twelve times; once every week in April, July and September, with 5 mm·m$^{-2}$ of water and 2.5, or 5 kg N·ha$^{-1}$ [40]. Water was added with a sprinkler kettle, irrigating over the canopy of *H. ammodendron*. The added N was $NH_4NO_3$ because the $NH_4^+$-N content was basically equal to the $NO_3^-$N content in the local atmospheric N deposition. To improve absorption of plants and better simulate atmospheric N deposition, we dissolved $NH_4NO_3$ in 0.5 L·m$^{-2}$ water and sprayed it to the plot uniformly with a sprayer. The control treatment (N0) was sprayed with the same amount of water. The time and frequency of adding N and water remained the same.

### Sampling of *H. ammodendron* and soil

Sampling was conducted in July of 2017. We collected the assimilating branches of *H. ammodendron* as our samples. Eight pieces of assimilating branches were obtained from each individual, with two pieces of assimilating branches being taken from each of the four cardinal directions with respect to the derection of full irradiance. All assimilating branches from the same plot were combined into a single sample. The topsoil (0–5 cm) of each treatment plot was also sampled. After sampling, the fresh soil samples were sieved through 2mm to remove the rocks and then air-dried in the laboratory.

### Measurements of N, P and K of *H. ammodendron*

The N content (N) of the assimilating branch samples was measured on a Delta[Plus] XP mass spectrometer (Thermo Scientific, Bremen, Germany) coupled with an automated elemental analyzer (Flash EA1112, CE Instruments, Wigan, UK) in the continuous flow mode. The standard deviation for the N measurements was 0.1%. The P and K content (P and K) of the assimilating branch samples was measured using an inductively coupled plasma source mass spectrometer (ICP-MS) after $HNO_3$-$H_2O_2$ digestion.

### Measurements of soil N, P contents and Olsen-P

Soil total N (STN) of each soil sample was measured using an automated elemental analyzer (VARIO EL cube). The standard deviation for the measurements was 0.05%. The soil total P (STP) of each soil sample was measured on a spectrophotometer by Mo-Sb colorimetry after $HClO$-$H_2SO_4$ digestion. In addition, we measured the soil Olsen-P (SOP) on a spectrophotometer by Mo-Sb colorimetry after $NaHCO_3$ extraction.

### Statistical analysis

Statistical analyses were conducted using the SPSS software (SPSS for Windows, Version 20.0, Chicago, IL, USA). One-way analysis of variance (ANOVA) and two-way ANOVA were used to compare the differences in plant N, P and K and soil N and P between each treatment. The least significant difference (LSD) test was conducted after each one-way ANOVA analysis to determine the difference in each index between every two treatments. Linear-regression analysis and Pearson analysis were used to determine the correlation between plants N, P, and K and between water use efficiency (WUE), including instantaneous WUE (ins-WUE) and intrinsic WUE (int-WUE), and plants N, P, K in *H. ammodendron* in all samples. Ins-WUE was calculated by the quotient of *A* and *E*, while int-WUE was calculated by the quotient of *A* and $g_s$. The data of WUE and gas exchange came from our published paper [46].

## Results

### Changes in plant N, P, and K and soil N, P across treatments

Plant N, P, and K concentrations in the assimilating branches ranged from 17.40 mg $g^{-1}$ to 30.68 mg $g^{-1}$, 0.73 mg $g^{-1}$ to 1.59 mg $g^{-1}$ and 13.98 mg $g^{-1}$ to 39.76 mg $g^{-1}$, respectively. One-way ANOVA analyses showed that plant N in W0N1 was significantly higher than that in other treatments (all p < 0.05 by LSD test, Fig 1a). Plant P was significantly higher in W0N1 than in W0N2 (p = 0.046), W1N0 (p = 0.035) and W1N1 (p = 0.017 by LSD test, Fig 1b). Plant K was significantly higher in W0N1 than in W0N2 (p = 0.006 by LSD test), W1N0 (p = 0.025 by LSD test), W1N1 (p = 0.001 by LSD test) and W1N2 (p = 0.043 by LSD test, Fig 1c). Two-way ANOVA analyses suggested that N addition played a role in plant N (Table 1). Water

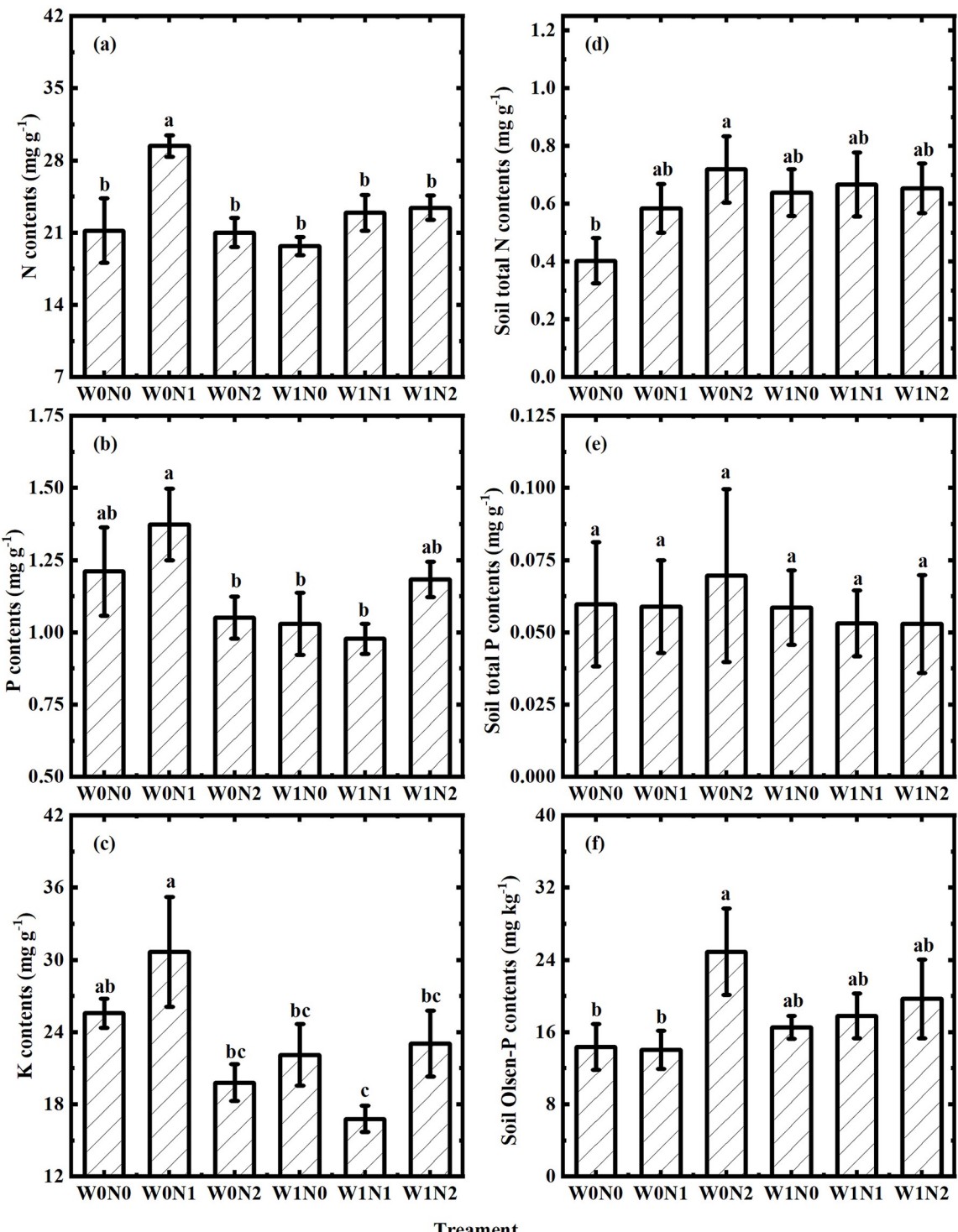

**Fig 1. Variations in plant N content (a), P content (b), K content (c), soil total N content (d), soil total P content (e) and soil Olsen-P content (f) across water (W) and nitrogen (N) additions.** The box represents the mean value of four replicates with error bars denoting the standard error (SE).

**Table 1. The p values of all measured and calculated indexes in plants under two-way ANOVA analysis of water (W) and nitrogen (N) additions.**

|  | W | N | W*N |
|---|---|---|---|
| N | 0.181 | 0.010* | 0.043* |
| P | 0.089 | 0.819 | 0.056 |
| K | 0.025* | 0.508 | 0.008** |
| STN | 0.298 | 0.180 | 0.153 |
| STP | 0.619 | 0.913 | 0.917 |
| SOP | 0.418 | 0.455 | 0.644 |

Note.

*, **indicates a significant influence.

addition affected plant K (Table 1). The interaction between water and N addition had an effect on plant N and K (Table 1).

STN, STP and SOP ranged from 0.23 mg g$^{-1}$ to 1.00 mg g$^{-1}$, 0.12 mg g$^{-1}$ to 1.29 mg g$^{-1}$ and 9.33 mg kg$^{-1}$ to 37.87 mg kg$^{-1}$, respectively. One-way ANOVA analyses showed that STN was significantly lower in W0N0 than in W0N2 ($p = 0.028$ by LSD test, Fig 1d). The STP did not change significantly across treatments (Fig 1e). However, SOP was significantly lower in W0N0 than in W0N2 ($p = 0.033$ by LSD test, Fig 1f). Two-way ANOVA analyses suggested that water addition, N addition, and their interactions had no significant effect on these three indexes (Table 1).

## Variations in the correlation between WUE and elements and the correlation between elements across water additions

For W0 treatments (including W0N0, W0N1 and W0N2), neither ins-WUE nor int-WUE were related to element content (S1 Table in S1 File). There were positive relationships between N and P (Fig 2a, Table 2) and between N and K (Fig 2b, Table 2). However, no correlation between P and K was observed (Fig 2c). For W1 treatments (including W1N0, W1N1 and W1N2), there was no correlation between WUE and element content (all $p > 0.05$, S1 Table in S1 File). Plant N was not related to P and K (Fig 2a and 2b), whereas K was positively correlated with P (Fig 2c, Table 2).

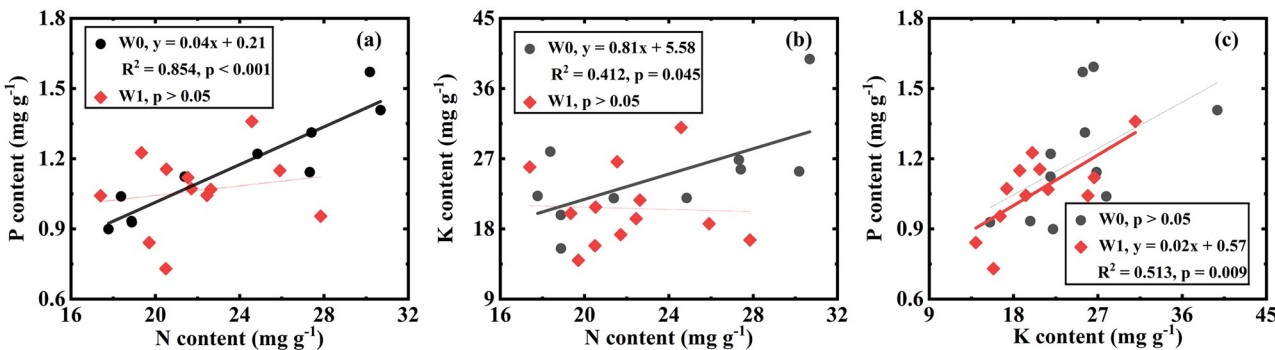

**Fig 2. Correlations of N vs. P (a), N vs. K (b), K vs. P (c) in W0 treatments (include W0N0, W0N1 and W0N2) and W1 treatments (include W1N0, W1N1 and W1N2).**

**Table 2. The slope of Linear-regression analysis among N, P and K.**

| Treatment | N vs. P | N vs. K | K vs. P |
|---|---|---|---|
| W0 | 0.040 | 0.811 | - |
| W1 | - | - | 0.024 |
| N0 | - | - | - |
| N1 | 0.046 | 1.533 | 0.022 |
| N2 | 0.041 | - | 0.027 |
| W0 and N | 0.043 | 1.257 | - |
| W1 and N | - | - | 0.027 |

Note: W0 treatment includes W0N0, W0N1 and W0N2 treatments, while W1 treatment included W1N0, W1N1 and W1N2 treatments. N0 treatment included W0N0 and W1N0 treatments, N1 treatment included W0N1 and W1N1treatments, while N2 treatment included W0N2 and W1N2 treatments. The W0 and N treatments included W0N1 and W0N2 treatments, whereas W1 and N treatments contained W1N1 and W1N2 treatments. Only the slopes that pass the significance test are shown in Table.

## Variations in the correlation between WUE and elements and the correlation between elements across N additions

For N0 treatments (including W0N0 and W1N0), there was no correlation between WUE and the elements, as well as no correlation between plant N, P and K (S1 Table in S1 File, Fig 3a–3c). For N1 treatments (including W0N1 and W1N1) also, WUE was not correlated with nutrients (S1 Table in S1 File), whereas N, P, and K were related to each other (Fig 3a–3c, Table 2). For N2 treatments (including W0N2 and W1N2) also, WUE was not correlated with element content (S1 Table in S1 File). Plant N was positively related to P (Fig 3a, Table 1), but not K (Fig 3b). Plant P was positively correlated with K (Fig 3c, Table 2).

## Variations in the WUE-element correlation and the N-P-K correlations across the interaction of water and N additions

Under the condition of adding N and no adding water (W0 + N treatments, including W0N1 and W0N2), WUE, including ins-WUE and int-WUE, was not related to element content (all p > 0.05, S1 Table in S1 File). N was positively correlated with P and K (Fig 4a and 4b, Table 2), and K had no coupling with P (Fig 4c). Under the treatments of adding both N and

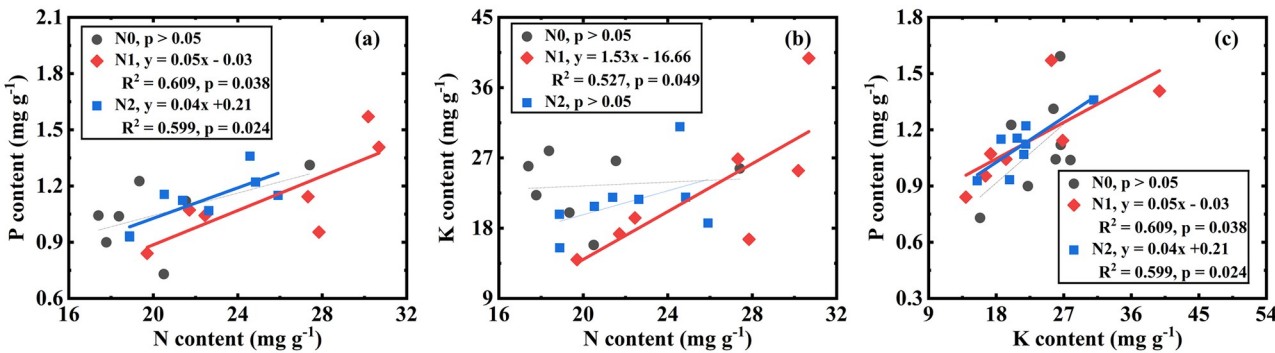

**Fig 3. Correlations of N vs. P (a), N vs. K (b), K vs. P (c) in N0 treatments (includes W0N0 and W1N0), N1 treatments (includeW0N1, and W1N1) and N2 treatments (include W0N2 and W1N2).**

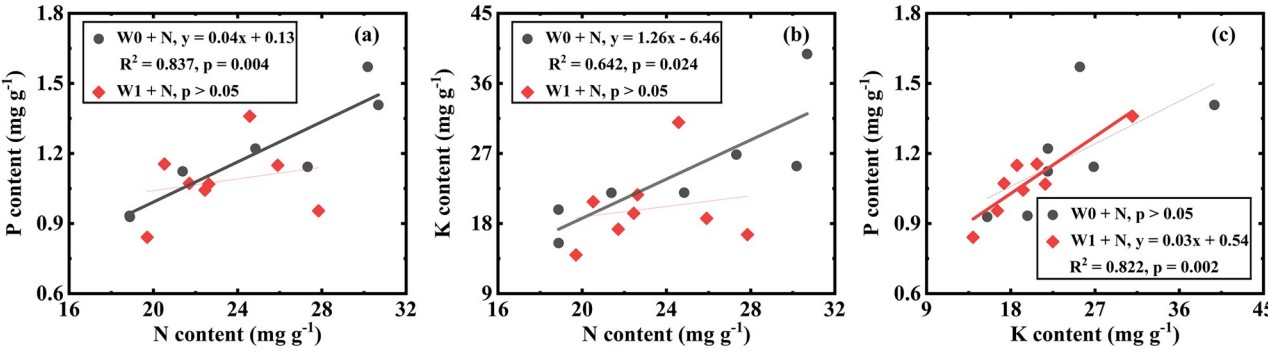

**Fig 4. Correlations of N vs. P (a), N vs. K (b), K vs. P (c) in W0 + N treatments (include W0N1 and W0N2) and N vs. P (d), N vs. K (e), K vs. P (f) in W1 + N treatments (include W1N1 and W1N2).**

water (W1 + N treatments, including W1N1 and W1N2), WUE, including ins-WUE and int-WUE, had no correlation with nutrients (S1 Table in S1 File). N was not related to P and K (Fig 4a and 4b), while K was positively correlated with P (Fig 4c, Table 2).

## Discussion

### The effect of changes in precipitation and N deposition on the correlation between WUE and elements in *H. ammodendron*

In the treatments with N or water alone, and in adding both N and water simultaneously, there was no significant correlation between WUE and N, P and K (S1 Table in S1 File). Thus, the result contradicted our hypothesis that there is a coupling between WUE and N, P and K in *H. ammodendron*. Although previous studies have observed the coupling between WUE and nutrients [6, 22–27], this study showed that this coupling is not a universal pattern. In addition, previous studies have suggested that the correlation between plant water economy and nutrition economy can be used to predict the physiological indicators of some plant that are difficult to monitor in real-time [47–49]. This prediction should not be suitable for *H. ammodendron*.

Gas exchange is the link between WUE and plants N, P, and K. Since ins-WUE is equal to $A$ divided $E$, ins-WUE should be positively related to $A$, and negatively related to $E$. In addition, given that int-WUE is calculated by the quotient of $A$ and $g_s$, int-WUE should have a positive relationship with $A$ and a negative correlation with $g_s$. As previously mentioned, plant N and P are associated with photosynthetic rate ($A$) [12–14, 16–18]; plant K strongly controls $g_s$ and $E$ [50]. In addition, $E$ and $g_s$ can regulate plant N, P, and K by affecting the absorption and transportation of these elements by plants. Therefore, plants N, P, and K have been observed to be related to gas exchange and WUE [6, 13, 16, 50]. However, there was no correlation between N, P, K, and gas exchange and no correlation between N, P, K, and WUE in *H. ammodendron* (S1, S2 Tables in S1 File). The lack of correlation between N, P, K and gas exchange may be related to the resource utilization strategy of *H. ammodendron*. The leaf economics spectrum (LES) defines a continuum spanning from fast-growing species with a rapid return of the investments in nutrients and carbon in leaves, to slow-growing species with a slower return [7]. Slow-growing species are characterized by a long life span, but low leaf nutrient concentrations, photosynthesis, respiration, and growth rates [47, 51–53]. *H. ammodendron* is a slow-growing species. It may be because of the slow return on the investment in nutrients that plant N, P, and K were not related to gas exchange. However, this hypothesis needs to be

confirmed. In addition, the water source of *H. ammodendron* could also cause a lack of correlation between N, P, K and gas exchange. The roots of *H. ammodendron* are exist at depth greater than 3 m into the soil to uptake groundwater [54]; therefore, groundwater is an important source of water for the plant species. The N, P, and K content in the groundwater may be relatively low. Therefore, even if gas exchange changes, the absorption of these elements by *H. ammodendron* does not necessarily change accordingly. This causes gas exchange to be independent of N, P, and K. Thus, there is no correlation between WUE and N, P, K.

## Effects of changes in precipitation and N deposition on the coupling between N, P and K in *H. ammodendron*

The correlation between plant N, P and K varied with water addition level (Fig 2), N addition level (Fig 3), and their interaction (Fig 4). These results confirmed our hypothesis that there is a relationship between N, P, and K for *H. ammodendron* and that the relationship is dependent on precipitation and N deposition.

An increase in rainfall promotes plant growth, leading to an increase in the demand for nutrients. However, in that case, plants may require more N and P than K because K is an important element for regulating osmotic pressure in plants. The increase in plant K reduces the water potential in plants, resulting in a decrease in water loss by transpiration. As a result, plants need to take up more K to resist drought stress [55–57]. Therefore, plant demand for K will decrease as precipitation increases. In contrast, the demand for P by *H. ammodendron* in the present study may be greater than the demand for N under increased precipitation. Ecologists use plant N/P ratio as an indicator of the relative limitation of N versus P in ecosystems, that is, N/P ratios less than 14 generally indicate N limitation, while N/P ratios greater than 16 suggest P limitation [58]. The N/P ratio in *H. ammodendron* was mostly higher than 16 (S3 Table in S1 File), suggesting that *H. ammodendron* suffered from P limitation. Thus, *H. ammodendron* will absorb more P relative to N and K as plant growth is enhanced by increasing precipitation. As a result, changes in precipitation disrupt the relationships between plant N, P and K.

Increased N deposition produces more N to the soil, thus promoting plant growth. As plant growth is promoted, it needs to absorb more N, P, and K. Due to the increase in soil N, plants absorb more N than P and K. As a result, increased N deposition disrupts the relationship between plants N, P, and K. Because both N deposition and precipitation affect the correlation between plant nutrients, it will inevitably lead to the influence of the interaction of N deposition and precipitation on this correlation.

Plants absorb N, P, and K in a specific proportion due to the proportionate requirement of N, P, and K to maintain optimal growth and development. The proportionate absorption of N, P, and K is a trade-off strategy, leading to the relationship between plant N, P, and K. However, this trade-off strategy may not be successful if the external environmental conditions change rapidly because N, P and K in natural ecosystems respond differently to changes in environmental conditions. N in natural ecosystems mainly originates from biological processes such as atmospheric N fixation and organic matter decomposition. The sources of P and K are mainly controlled by geochemical processes such as rock weathering. Therefore, changes in environmental conditions cause different changes in the availability of soil N, P, and K. Consequently, the plant's acquisition of N, P, and K will also change, resulting in disproportionate absorption of nutrients, which is another trade-off strategy. Thus, our results suggest that global changes in precipitation and N deposition influence the trade-off strategy for N, P, and K absorption in *H. ammodendron*. However, it should be highlighted that the trade-off strategy of the disproportionate absorption of N, P, and K is temporary. If precipitation and N input

continue to increase for a long time, plants will eventually return to a state of absorbing nutrients proportionately to survive.

The absorption and utilization of elements by plants is an important process in the biogeochemical cycles of terrestrial ecosystems, and it can affect the biogeochemical cycles of elements in terrestrial ecosystems. This study observed that N, P, and K in *H. ammodendron* and the relationship between N, P, and K changed with N deposition and precipitation, suggesting that the biogeochemical cycles of N, P, and K in desert ecosystems dominated by *H. ammodendron* would be affected by N deposition and precipitation.

## Conclusion

In any water or nitrogen treatment, the WUE remained independent of N, P, and K of *H. ammodendron*, suggesting that there is no coupling between water use and nutrient economics in the dominant species of the Asian desert. This result was associated with lack of the correlation between N, P, K, and gas exchange in *H. ammodendron*. However, water addition, N addition, and the interaction between them exerted effects on the relationships between plant N, P, and K. This indicates that global changes in precipitation and atmospheric N deposition will affect the geochemical cycles of N, P, and K in Asian deserts dominated by *H. ammodendron*, and drive the relationships between plant nutrients to change, thereby causing changes in the trade-off strategy of *H. ammodendron*.

## Supporting information

**S1 File. Supporting tables—Contains all the supporting tables.**
(DOCX)

**S1 Dataset. The N, P, K contents of *H. ammodendron*.**
(XLSX)

**S2 Dataset. The photosynthetic rate (A), stomatal conductance (gs) and transpiration rate (E) in *H. ammodendron*.**
(XLSX)

**S3 Dataset. The soil total N (STN), soil total P (STP) and soil Olsen-P (SOP).**
(XLSX)

## Acknowledgments

We would like to thank the supports from the Fukang Observation Station of Desert Ecology, Xinjiang Institute of Ecology and Geography, Chinese Academy of Sciences. We would like to thank Dr. Eric Posmentier for his English editing on this article.

## Author Contributions

**Formal analysis:** Xuejun Liu, Guoan Wang.

**Funding acquisition:** Guoan Wang.

**Investigation:** Zixun Chen, Xuejun Liu, Xiaoqing Cui, Yaowen Han.

**Methodology:** Guoan Wang.

**Resources:** Xuejun Liu, Guoan Wang.

**Writing – original draft:** Zixun Chen.

**Writing – review & editing:** Zixun Chen, Guoan Wang.

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
