## [Decision Letter · Decision Letter 0]

20 Apr 2021

PONE-D-21-06220

Precipitation change and atmospheric N deposition affect the correlation between N, P and K but not the coupling of water-element in H. ammodondron

PLOS ONE

Dear Dr. Wang,

Thank you for submitting your manuscript to PLOS ONE. After careful consideration, we feel that it has merit but does not fully meet PLOS ONE’s publication criteria as it currently stands. Therefore, we invite you to submit a revised version of the manuscript that addresses the points raised during the review process.

The authors need to modify the manuscript carefully based on the comments of two reviewers. Please write the results and discussion separately. Then you need language editing by a native speaker.

We look forward to receiving your revised manuscript.

Kind regards,

Yajuan Zhu, Ph.D.

Academic Editor

PLOS ONE

Journal Requirements:

PLOS requires an ORCID iD for the corresponding author in Editorial Manager on papers submitted after December 6th, 2016. Please ensure that you have an ORCID iD and that it is validated in Editorial Manager. To do this, go to ‘Update my Information’ (in the upper left-hand corner of the main menu), and click on the Fetch/Validate link next to the ORCID field. This will take you to the ORCID site and allow you to create a new iD or authenticate a pre-existing iD in Editorial Manager. Please see the following video for instructions on linking an ORCID iD to your Editorial Manager account: https://www.youtube.com/watch?v=_xcclfuvtxQ

Additional Editor Comments:

Please modify the manuscript carefully and answer questions from two reviewers. We are expecting the revised mansucript.

Reviewers' comments:

Reviewer's Responses to Questions

**Comments to the Author**

1. Is the manuscript technically sound, and do the data support the conclusions?

Reviewer #1: Yes

Reviewer #2: Yes

2. Has the statistical analysis been performed appropriately and rigorously? 

Reviewer #1: Yes

Reviewer #2: Yes

3. Have the authors made all data underlying the findings in their manuscript fully available?

Reviewer #1: Yes

Reviewer #2: Yes

4. Is the manuscript presented in an intelligible fashion and written in standard English?

Reviewer #1: Yes

Reviewer #2: Yes

5. Review Comments to the Author

Reviewer #1: The manuscript “Precipitation change and atmospheric N deposition affect the correlation between N, P and K but not the coupling of water-element in H. ammodondron” by Chen et al investigated the effects from precipitation change and N deposition on the relationships between different plant nutrient (N, P and K) contents and water-use efficiency of Haloxylon ammodendron, which is a dominant desert plant. The authors showed that addition of water and N altered the correlations between nutrient contents in plant but not the correlations between plant WUE and nutrient contents. This result is interesting because it suggests global change factors such as precipitation change and N deposition can influence terrestrial biogeochemical cycles by affecting plant water-nutrient trade-off strategies. However, the manuscript needs to be properly revised before the acceptance for publication, especially for the language. I will not specify the mistakes in English writing because there were too many. Thus a thorough revision by native English speaker is strongly recommended.

Title: I think the name of the studied plant should be in full spell.

Introduction:

L55 “…, which are affected by plant N…” I don’t quite understand what did this “which” refer to, WUE or gas exchanges? Also for this sentence L55-57 I think the authors should explain further why the WUE (or water use) is expected to be related to plant nutrient contents. This is the basis and significance of understanding the results of this study. I see that the authors tried to bring this issue in L60-62, but the explanation was not clear.

L66-68 Here the authors mentioned that “little attention has been paid to the influence of N deposition and precipitation change on these relationships”. However in the next paragraph, “the effects of rainfall change and N deposition on these couplings have been observed in grassland and forest ecosystems [22,23]” (L77-79). I think the current progresses on this topic should be properly introduced in the section.

Overall this section lacks a clear review on the current knowledge of the coupling between plant WUE and nutrient status (related to absorption and utilization) and its response to global change factors.

Results:

In the experiment there were two factors, that is, water addition and N addition. However, the authors only used ANOVA to analyze the effects of different treatments on plant WUE, nutrients and soil variables. As such, the effects of watering, N addition and their interactions on these variables were not statistically assessed. In addition, the authors used a lot of “slightly” or “slight” in describing the changes in certain variables by treatments of watering or N addition. This is not appropriate because it cannot reflect the significance of the statistical tests. The description of results should be precise about whether the change is significant or not. Words like “slightly higher” should be revised unless the authors feel a strong need to describe the result that way.

L223-225 It is my feeling that this subtitle, as well as the next ones in L252-253 and L273-275 are too long.

L273-275 This subtitle is confusing. Does it means that here the authors want to present the correlations of data from all treatments?

Also the figures and their captions can be improved. I understand that the authors are trying to present the different correlations of N-P (and N-K and P-K) as affected by watering or N addition by comparing the slopes (or significance) of the linear regressions. It would be better to present the linear regression lines in a same plot, which may provide a more direct view on the effect of watering or N addition level. For example in Figure 2, for the N-P relationship, the authors may put data from W0 and W1 treatments all in one plot but with different colors, which may help to show the different N-P relationships as affected by watering level.

Discussions:

L308-321 I feel that it is better to put these sentences in the Introduction section. Some other sentences in this paragraph are also useful in the Introduction as information of the relationship between plant WUE and nutrient status.

L337-345 Here the authors proposed that the lack of significant coupling between WUE and nutrient contents is related to the slow-growing of H. ammodendron plant. I wonder if there is any special characteristic in leaf or root morphology of this species that helps to survive the desert. These characteristics may also be relevant for the uncoupled plant WUE and nutrient status.

L362-366 I would like to see a more precise explanation on the mechanisms of the trade-off strategies that result in the “proportional” and “disproportional” absorption of nutrients by plant.

L394-398 This sentence is too long and confusing.

Overall I feel the authors discussed too many on the changes in plant NPK contents with watering and N addition and used it as the explanation of the varied NPK coupling as affected by watering and N addition. There are many literatures on the ecosystem stoichiometry including plant nutrient ratios. I suggest that the authors should consider the underlying mechanisms of theses plant stoichiometric ratios, and make efforts to explain their results based on both plant physiology and ecology aspects.

Conclusion:

L414-415 “… all exerted significant effects …” From what I understand, this statement is based on comparing the slope of linear regressions (or correlation coefficients) under different watering or N addition levels. I don’t see a statistical analysis on the effect of watering, N addition and their interactions on the correlations. So the word “significant” can be misleading.

Reviewer #2: The manuscript titled: “Precipitation change and atmospheric N deposition affect the correlation between N, P and K but not the coupling of water-element in H. ammodondron” by Chen et al. presents interesting results on the Gurbantunggut Desert. But there are a few significant limitations that need to be addressed in a revised version of this manuscript.

1. The whole paper needs a thorough edit (and spelling check) by an English first-language speaker before resubmission.

2. Results and Discussion: Your paper has two sections such as “Results and discussion”, so you should just represent the results with concise and precise sentences and then take further discussion in the discussion section. But in your paper, lots of discussion sentences appear your results section. Please cancel all information related to discussion in your results section

6. PLOS authors have the option to publish the peer review history of their article (what does this mean?). If published, this will include your full peer review and any attached files.

Reviewer #1: No

Reviewer #2: No

---

## [Author Response · Author response to Decision Letter 0]

12 May 2021

Reviewer 1

Title: I think the name of the studied plant should be in full spell.

Response:

Thank you for your suggestion! We have revised the title. Please see line 3 at the Revised Manuscript with Track Changes.

Introduction:

L55 “…, which are affected by plant N…” I don’t quite understand what did this “which” refer to, WUE or gas exchanges? Also for this sentence L55-57 I think the authors should explain further why the WUE (or water use) is expected to be related to plant nutrient contents. This is the basis and significance of understanding the results of this study. I see that the authors tried to bring this issue in L60-62, but the explanation was not clear.

Response:

Thank you for your comment! We have revised our writing and added the reason why WUE is expected to be related to plant N, P and K. Please see line 56-68 at the Revised Manuscript with Track Changes.

L66-68 Here the authors mentioned that “little attention has been paid to the influence of N deposition and precipitation change on these relationships”. However in the next paragraph, “the effects of rainfall change and N deposition on these couplings have been observed in grassland and forest ecosystems [22,23]” (L77-79). I think the current progresses on this topic should be properly introduced in the section.

Response:

Sorry! Our writing may mislead you. From line 74 to line 108 at the Revised Manuscript with Track Changes, we first put forward the main concern in our article, which is also the problem that this article will solve. Then around this concern, we made 3 hypotheses. (1) The main concern is that little attention has been paid to the influence of N deposition and precipitation change on these relationships between WUE and plant N, P, K, thus, there is great uncertainty about this issue; although previous studies have reported that N deposition and precipitation change have resulted in the variation of plant N/P ratio and N/K ratio, which indirectly confirmed that plant element coupling was affected by the changes in rainfall and N deposition (see lines 74-89). (2) The coupling between WUE and plant N, P, K and the coupling between plant elements have been found in grassland and forest ecosystems, we made the first hypothesis that these couplings should also occur in H. ammodendron (see lines 96-100). (3) The effects of rainfall change and N deposition on these element couplings have been observed in two meta-analyses, thus, we made the second hypothesis that the correlation between plant nutrients should also vary with changes in rainfall and atmospheric N deposition for H. ammodendron (see lines 100-104). (4) The respond of elements to N deposition and precipitation may vary across elements, and WUE is influenced by N deposition and precipitation; therefore we made the third hypothesis that changes in rainfall and atmospheric N deposition also have an effect on these correlations between WUE and N, P, K (see lines 104-108).

Overall this section lacks a clear review on the current knowledge of the coupling between plant WUE and nutrient status (related to absorption and utilization) and its response to global change factors.

Response:

Thank you for your comment! We have added the review on the current knowledge of the coupling between plant WUE and nutrient status (related to absorption and utilization) and its response to global change factors, especially precipitation and atmospheric N deposition. Please see the Introduction section (Lines 56-68, 80-89 at the Revised Manuscript with Track Changes).

Results:

In the experiment there were two factors, that is, water addition and N addition. However, the authors only used ANOVA to analyze the effects of different treatments on plant WUE, nutrients and soil variables. As such, the effects of watering, N addition and their interactions on these variables were not statistically assessed. In addition, the authors used a lot of “slightly” or “slight” in describing the changes in certain variables by treatments of watering or N addition. This is not appropriate because it cannot reflect the significance of the statistical tests. The description of results should be precise about whether the change is significant or not. Words like “slightly higher” should be revised unless the authors feel a strong need to describe the result that way.

Response:

Thank you for your suggestions. Based on these suggestions, we have made a big change. We have conducted a two-way ANOVA analysis to assess the effects of watering, N addition and their interactions on plant N, P, K and soil N, P. Please see lines 216-233 and Table 1 at the Revised Manuscript with Track Changes. In addition, we have simplified the expression of the results, and deleted many verbose and meaningless expressions.

L223-225 It is my feeling that this subtitle, as well as the next ones in L252-253 and L273-275 are too long.

Response:

Thank you for your comment. We have revised these subtitles. Please see lines 244-246, 273-275 and 297-299 at the Revised Manuscript with Track Changes.

L273-275 This subtitle is confusing. Does it means that here the authors want to present the correlations of data from all treatments?

Response:

Sorry for our poor writing. In this sub-section, we presented the correlation between WUE and N, P, K and the correlation between plant N, P, K under two conditions: adding N and no adding water (W0 + N treatments, includes W0N1 and W0N2) and adding both N and water (W1 + N treatments, includes W1N1 and W1N2). The aim is to assess the effect of the interaction of water and N addition on the correlation between WUE and N, P, K and the correlation between plant N, P, K. We have revised this subtitle. Please see lines 297-299 at the Revised Manuscript with Track Changes.

Also the figures and their captions can be improved. I understand that the authors are trying to present the different correlations of N-P (and N-K and P-K) as affected by watering or N addition by comparing the slopes (or significance) of the linear regressions. It would be better to present the linear regression lines in a same plot, which may provide a more direct view on the effect of watering or N addition level. For example in Figure 2, for the N-P relationship, the authors may put data from W0 and W1 treatments all in one plot but with different colors, which may help to show the different N-P relationships as affected by watering level.

Response:

Thank you for your suggestion. We have revised the figure, please see Fig. 2-4.

Discussions:

L308-321 I feel that it is better to put these sentences in the Introduction section. Some other sentences in this paragraph are also useful in the Introduction as information of the relationship between plant WUE and nutrient status.

Response:

Thank you for your comment. We have moved these sentences into the Introduction. Please see lines 56-68 at the Revised Manuscript with Track Changes.

L337-345 Here the authors proposed that the lack of significant coupling between WUE and nutrient contents is related to the slow-growing of H. ammodendron plant. I wonder if there is any special characteristic in leaf or root morphology of this species that helps to survive the desert. These characteristics may also be relevant for the uncoupled plant WUE and nutrient status.

Response:

Your idea is correct. It may be associated with the roots of H. ammodendron. They are inserted into the soil layer deeper than 3 m to uptake groundwater (Sheng et al., 2004), so groundwater is an important or most important source of water for the plant species. The N, P, K contents in groundwater may be relatively low. Therefore, even if gas exchanges change, the absorption of these elements by H. ammodendron does not necessarily change accordingly, resulting in gas exchanges independent of N, P, K and thus no correlation between WUE and N, P, K. We added the possible mechanism. please see lines 353-360 at the Revised Manuscript with Track Changes.

L362-366 I would like to see a more precise explanation on the mechanisms of the trade-off strategies that result in the “proportional” and “disproportional” absorption of nutrients by plant.

Response:

Thank you for your comment. The proportional absorption is caused by the proportional requirement of N, P, K by plants. The disproportional absorption may be associated with disproportional availabilities of soil N, P, K induced by adding water and adding N. However, the trade-off strategies of disproportional absorption of N, P and K is temporary. If precipitation and N input increase for a long time, in order to survive, plants will eventually return to a state of absorbing nutrients in proportion. We have added the detailed explanation, please see lines 396-407 at the Revised Manuscript with Track Changes.

L394-398 This sentence is too long and confusing.

Response:

Thank you for your comment. Since this sentence is verbose and meaningless, we have deleted it. Furthermore, we have deleted S1 Table and S3 Table because these two tables are also meaningless.

Overall I feel the authors discussed too many on the changes in plant NPK contents with watering and N addition and used it as the explanation of the varied NPK coupling as affected by watering and N addition. There are many literatures on the ecosystem stoichiometry including plant nutrient ratios. I suggest that the authors should consider the underlying mechanisms of theses plant stoichiometric ratios, and make efforts to explain their results based on both plant physiology and ecology aspects.

Response:

Thank you for your suggestion. Based on the suggestion, we made a great revision on the discussion (see lines 374-395 at the Revised Manuscript with Track Changes). We used the knowledges of the physiology of K (lines 376-380) and the ecology theory of P limitation (lines 380-387) to explain our results. In addition, in order to prove that H. ammodendron suffered from P limitation, we have added a new S3 Table showing the N/P ratio in H. ammodendron.

Conclusion:

L414-415 “… all exerted significant effects …” From what I understand, this statement is based on comparing the slope of linear regressions (or correlation coefficients) under different watering or N addition levels. I don’t see a statistical analysis on the effect of watering, N addition and their interactions on the correlations. So the word “significant” can be misleading.

Response:

Thank you for your suggestion. We have deleted this word. Please see line 428 at the Revised Manuscript with Track Changes.

Reviewer 2

1. The whole paper needs a thorough edit (and spelling check) by an English first-language speaker before resubmission. 

Response:

Thank you for your suggestion. We have asked Dr. Eric Posmentier in the department of Earth Sciences, Dartmouth College to edit English. 

2. Results and Discussion: Your paper has two sections such as “Results and discussion”, so you should just represent the results with concise and precise sentences and then take further discussion in the discussion section. But in your paper, lots of discussion sentences appear your results section. Please cancel all information related to discussion in your results section

Response:

Thank you for your suggestion. we have simplified the expression of the results, and deleted many verbose and meaningless expressions and the discussion in the Results Section. Please see lines 216-233, 255-257, 286-289 and 310-312 at the Revised Manuscript with Track Changes.

3. Specific comments：

Line 49 water is the Plant water economy and nutrition economy? please define it clearly, and early, in the introduction.

Response:

Thank you for your suggestion. We have revised our writing. Please see lines 50-51 at the Revised Manuscript with Track Changes.

Line 55-56 I think references are needed here.

Response:

Thank you for your suggestion. We have added a description of the mechanism behind the correlation between WUE and gas exchange and N, P, K; and references have been cited in this part. Please see lines 56-68 at the Revised Manuscript with Track Changes.

Line 77-81 is there a direct logical relationship between the two ecosysytems?

Response:

Thank you for your comment. We made a mistake in the previous version. These two previous studies [32,33] (the original version is [22, 23]) are meta-analysis on global scale, not research done in a single ecosystem. We have revised the expression, please see lines 100-102 at the Revised Manuscript with Track Changes. Since the effects of changes in rainfall change and N deposition on the N, P, K couplings have been observed on global scale, we assumed that the correlation between plant nutrients should also vary with changes in rainfall and atmospheric N deposition for H. ammodendron. This is the logic of our writing.

Line 106 suggest to revises “woods” to “trees”

Response:

Thank you for your suggestion. We have revised this word. Please see line 133 at the Revised Manuscript with Track Changes.

Line 136 suggest to delete “Due to no leaves,” I think this is an inaccurate statement.

Response:

Thank you for your suggestion. We have revised our writing. Please see line 163 at the Revised Manuscript with Track Changes.

Line 156 suggest to delete “which was its prime assimilating organ”

Response:

Thank you for your suggestion. We have deleted this sentence. Please see line 184 at the Revised Manuscript with Track Changes.

Line 156 what is “slight positive”? It is unclear

Response:

Sorry. I don’t see “slight positive” in line 156. However, due to this expression is unclear and useless, we have deleted this expression in the whole manuscripts.

Line 156 How to understand “no relationships between…….” I think this is an inaccurate statement

Response:

Thank you for your suggestion. We have revised our writing. Please see line 251 at the Revised Manuscript with Track Changes.

Line 264-266 Please, check the language in this statement. It is unclear

Response:

Thank you for your suggestion. Since this sentence may be inapposite. We have deleted it. Please see lines 286-289 at the Revised Manuscript with Track Changes.

---

## [Decision Letter · Decision Letter 1]

4 Jun 2021

PONE-D-21-06220R1

Changes in Precipitation and atmospheric N deposition affect the correlation between N, P and K but not the coupling of water-element in Haloxylon ammodondron

PLOS ONE

Dear Dr. Wang,

Thank you for submitting your manuscript to PLOS ONE. After careful consideration, we feel that it has merit but does not fully meet PLOS ONE’s publication criteria as it currently stands. Therefore, we invite you to submit a revised version of the manuscript that addresses the points raised during the review process.

Please check your manuscript carefully, especially semicolon. The reviewer  has given details on format mistakes.

We look forward to receiving your revised manuscript.

Kind regards,

Yajuan Zhu, Ph.D.

Academic Editor

PLOS ONE

Journal Requirements:

Additional Editor Comments (if provided):

Please check your manuscript carefully, especially semicolon. The reviewer has give details on format mistakes.

Reviewers' comments:

Reviewer's Responses to Questions

**Comments to the Author**

1. If the authors have adequately addressed your comments raised in a previous round of review and you feel that this manuscript is now acceptable for publication, you may indicate that here to bypass the “Comments to the Author” section, enter your conflict of interest statement in the “Confidential to Editor” section, and submit your "Accept" recommendation.

Reviewer #1: All comments have been addressed

Reviewer #2: All comments have been addressed

2. Is the manuscript technically sound, and do the data support the conclusions?

Reviewer #1: Yes

Reviewer #2: Yes

3. Has the statistical analysis been performed appropriately and rigorously? 

Reviewer #1: Yes

Reviewer #2: Yes

4. Have the authors made all data underlying the findings in their manuscript fully available?

Reviewer #1: Yes

Reviewer #2: Yes

5. Is the manuscript presented in an intelligible fashion and written in standard English?

Reviewer #1: Yes

Reviewer #2: Yes

6. Review Comments to the Author

Reviewer #1: I see that the manuscript was greatly improved from the old version. There are still some problems, but most of them are minor and easy to correct. In particular I suggest the authors to take care of the use of semicolon. There were quite a lot in the manuscript, which made some sentences too long to read.

Here are my specific comments on the ms:

L52: The first appearance of N, P, and K should be in their full spell. In addition, it is better to use their full spell if they are the first words in a sentence, e.g. L61 and L64.

L65: change to “osmotic pressure and enzyme activity [19-21], and thus strongly controls …”

L74-77: The first sentence is the same with L48-49, so it is not necessary to state again. Please delete these texts before the semicolon. In addition, I noticed that the authors used quite a lot of semicolons in the text, which has caused the manuscript to be less readable. Please revise.

L82-84: Here is an example of the bad use of semicolon. The sentence should be separated into two sentences for better reading.

L88-89: Another semicolon again. In addition, I feel that the two parts have stated the similar meaning. The influence of precipitation and N deposition change on WUE-nutrient relationship is less known, so of course there is uncertainty. But what does this uncertainty mean or imply? I suggest the authors should expand further to explain this uncertainty, otherwise the latter sentence after the semicolon is useless and should be deleted.

L96-108: I see that the authors are trying to explain how did these hypotheses come out, but the texts are in fact a repeated of those in the above paragraphs. I suggest to revise these sentences into “Given the universal plant WUE-nutrient coupling and plant NPK coupling, and the possible impact of changes in precipitation and N deposition on these couplings, we hypothesized that: (1)…; (2)…; (3)….”

L108: change to “we conduced a field experiment …”

L112: “grown” should be “growing”

L113: move “in desert ecosystem” to the end of this sentence.

L217: Please check if this is the standard form of unit by PLOS publication.

L218-223: More semicolons. Why use the semicolon when the sentence can be finished with a period? Also, I don’t follow the use of p-value here and the next paragraph with STN and SOP. Does the p-value represent t-test or LSD comparison of two treatments? If so, this should be clearly stated either here or in the statistical analysis method section. There should be only one p-value for the ANOVA with six treatments.

L223-226: The same problem about semicolon. In addition, these p-values are actually presented in the table, so it is not necessary to repeatedly present in the text. The same problem also happened in the following sections of results, please check and revise.

L230: Change to “STP did not change significantly across treatments”, and delete “p>0.05”. Same for the last sentence and elsewhere in the manuscript.

Also, I wonder why the data for WUE under different treatments were not provided. Did WUE change significantly with N addition and watering?

L247-251: Still the semicolon problem. The text clearly presented two sets of the results, i.e. the WUE-NPK correlations and the NPK correlations, so why not separate them with a period? Also details of the linear regressions were provided in the figure and again in the Table 2, and once again in the text here. Please only present what you think the most important information in the parentheses and delete the repeated ones. The same problem with the next sentence and sentences in the next two paragraphs.

L323: change to “there was no significant correlation”

L325-327: This sentence means the same with L322-324. Please delete it.

L341-343: This sentence described a result (Table S2), which should be mentioned in the Results. Overall the results of WUE (or gas exchange parameters) were lost from the Results. Does the authors have a good reason for not presenting these results?

L348: change “and” to “but”

L352: change to “needs confirmation” or “needs to be confirmed”

L353: There is no need to start a new paragraph because here presented another reason for the non-significant correlation between WUE and plant NPK.

L364-370: These three sentences can be simplified into one sentence. For example: “The correlations between plant NPK varied with water addition levels (Fig. 2), N addition levels (Fig. 3), as well as the interaction between N addition and watering (Fig. 4). There results confirm …”

L375: change to “in that case”

L380-381: change “on the other hand” to “in contrast”. Also is there any reference or explanation on this description that plant demands for P is greater than that for N when precipitation increases (the authors did not point out which direction of the change in precipitation)?

L382-384: Bad grammar. Change to “… indicator of the relative limitation of N versus P in ecosystems, i.e., N/P ratio less than 14 generally indicates N limitation while N/P ratio greater than 16 suggests P limitation.”

L384-385: “almost higher than 16”? Do you mean “mostly higher than 16”? From Table S3, all mean N/P ratios were higher than 16. Also this should be mentioned in the Results section.

L394: Where did the “interaction of temperature and precipitation” come from?

Reviewer #2: This is the second time I have reviewed this manuscript, and I feel the authors have really improved the manuscript and limited the scope of their results to accurately reflect what the data are indicating.but,the standard of English is poor - even in this revision. Providing this is done, the manuscript is in my opinion acceptable for publication.

7. PLOS authors have the option to publish the peer review history of their article (what does this mean?). If published, this will include your full peer review and any attached files.

Reviewer #1: No

Reviewer #2: No

---

## [Author Response · Author response to Decision Letter 1]

21 Jun 2021

Reviewer #1: I see that the manuscript was greatly improved from the old version. There are still some problems, but most of them are minor and easy to correct. In particular I suggest the authors to take care of the use of semicolon. There were quite a lot in the manuscript, which made some sentences too long to read.

Response:

Thank you for your suggestion! We have reduced the use of semicolon.

Here are my specific comments on the ms:

L52: The first appearance of N, P, and K should be in their full spell. In addition, it is better to use their full spell if they are the first words in a sentence, e.g. L61 and L64.

Response:

Thank you for your suggestion! We have revised the spell. Please see lines 54, 55, 57, 65 and 68 at the Revised Manuscript with Track Changes.

L65: change to “osmotic pressure and enzyme activity [19-21], and thus strongly controls …”

Response:

Thank you for your suggestion! We have revised the sentence. Please see lines 69-70 at the Revised Manuscript with Track Changes.

L74-77: The first sentence is the same with L48-49, so it is not necessary to state again. Please delete these texts before the semicolon. In addition, I noticed that the authors used quite a lot of semicolons in the text, which has caused the manuscript to be less readable. Please revise.

Response:

Thank you for your suggestion! We have revised the sentence. Please see lines 79-81 at the Revised Manuscript with Track Changes. And we have reduced the use of semicolon as far as possible.

L82-84: Here is an example of the bad use of semicolon. The sentence should be separated into two sentences for better reading.

Response:

Thank you for your suggestion! We have revised the sentence. Please see lines 88-90 at the Revised Manuscript with Track Changes.

L88-89: Another semicolon again. In addition, I feel that the two parts have stated the similar meaning. The influence of precipitation and N deposition change on WUE-nutrient relationship is less known, so of course there is uncertainty. But what does this uncertainty mean or imply? I suggest the authors should expand further to explain this uncertainty, otherwise the latter sentence after the semicolon is useless and should be deleted.

Response:

Thank you for your suggestion! We have revised the sentence. Please see line 96 at the Revised Manuscript with Track Changes.

L96-108: I see that the authors are trying to explain how did these hypotheses come out, but the texts are in fact a repeated of those in the above paragraphs. I suggest to revise these sentences into “Given the universal plant WUE-nutrient coupling and plant NPK coupling, and the possible impact of changes in precipitation and N deposition on these couplings, we hypothesized that: (1)…; (2)…; (3)….”

Response:

Thank you for your suggestion! We have revised the sentence. Please see lines 104-111 at the Revised Manuscript with Track Changes.

L108: change to “we conduced a field experiment …”

Response:

Thank you for your suggestion! We have revised the sentence. Please see line 122 at the Revised Manuscript with Track Changes.

L112: “grown” should be “growing”

Response:

Thank you for your suggestion! We have revised the word. Please see line 127 at the Revised Manuscript with Track Changes.

L113: move “in desert ecosystem” to the end of this sentence.

Response:

Thank you for your suggestion! We have revised the sentence. Please see lines 128-129 at the Revised Manuscript with Track Changes.

L217: Please check if this is the standard form of unit by PLOS publication.

Response:

Thank you for your suggestion! We have revised the unit. Please see lines 243-244 and 255-256 at the Revised Manuscript with Track Changes.

L218-223: More semicolons. Why use the semicolon when the sentence can be finished with a period? Also, I don’t follow the use of p-value here and the next paragraph with STN and SOP. Does the p-value represent t-test or LSD comparison of two treatments? If so, this should be clearly stated either here or in the statistical analysis method section. There should be only one p-value for the ANOVA with six treatments.

Response:

Thank you for your suggestion! We have deleted the semicolon. Please see lines 244-250 at the Revised Manuscript with Track Changes. The p-value represent LSD comparison, we have added the statements. Please see lines 230-232, 246, 248-250, and 258-260.

L223-226: The same problem about semicolon. In addition, these p-values are actually presented in the table, so it is not necessary to repeatedly present in the text. The same problem also happened in the following sections of results, please check and revise.

Response:

Thank you for your suggestion! We have deleted the semicolon and the redundant p-values. Please see lines 250-254 at the Revised Manuscript with Track Changes.

L230: Change to “STP did not change significantly across treatments”, and delete “p>0.05”. Same for the last sentence and elsewhere in the manuscript.

Response:

Thank you for your suggestion! We have revised the sentence. Please see lines 258-259 and 262 at the Revised Manuscript with Track Changes.

Also, I wonder why the data for WUE under different treatments were not provided. Did WUE change significantly with N addition and watering?

Response:

The data have been published in another paper (Chen et al., 2021, Biogeosciences, 18, 2859–2870). In addition, the main objective of the present study is to determine the effect of precipitation and atmospheric N deposition on the correlation between N, P and K and between WUE and N, P, K in H. ammodendron. Therefore, the data for WUE under different treatments has not been provided in this manuscript. We have deleted the description of WUE and gas exchange measurement in the Materials and methods Section, and added a statement about the data source of WUE and gas exchange in the manuscript. Please see lines 236-238 at the Revised Manuscript with Track Changes. We have reported in the paper published in BG that WUE changed significantly with N addition and watering. 

L247-251: Still the semicolon problem. The text clearly presented two sets of the results, i.e. the WUE-NPK correlations and the NPK correlations, so why not separate them with a period? Also details of the linear regressions were provided in the figure and again in the Table 2, and once again in the text here. Please only present what you think the most important information in the parentheses and delete the repeated ones. The same problem with the next sentence and sentences in the next two paragraphs.

Response:

Thank you for your suggestion! We have revised the sentence and deleted the redundant information in the parentheses. Please see lines 275-283 at the Revised Manuscript with Track Changes.

L323: change to “there was no significant correlation”

Response:

Thank you for your suggestion! We have revised the sentence. Please see line 337 at the Revised Manuscript with Track Changes.

L325-327: This sentence means the same with L322-324. Please delete it.

Response:

Thank you for your suggestion! We have deleted the sentence. Please see line 340-341 at the Revised Manuscript with Track Changes.

L341-343: This sentence described a result (Table S2), which should be mentioned in the Results. Overall the results of WUE (or gas exchange parameters) were lost from the Results. Does the authors have a good reason for not presenting these results?

Response:

Thank you for your suggestion! However, the aim of the present study is to determine the effect of precipitation and atmospheric N deposition on the correlation between N, P and K and between WUE and N, P, K in H. ammodondron. In addition, the results of gas exchanges and WUE have been reported in our paper published in Biogeosciences (Chen et al., 2021, Biogeosciences, 18, 2859–2870). We presented the results of the correlation between N, P, K and gas exchange in S2 Tables to only explain the observed irrelevance between N, P, K and WUE. So we did not present them in the Results section. 

L348: change “and” to “but”

Response:

Thank you for your suggestion! We have revised the word. Please see line 364 at the Revised Manuscript with Track Changes.

L352: change to “needs confirmation” or “needs to be confirmed”

Response:

Thank you for your suggestion! We have revised the sentence. Please see lines 367-368 at the Revised Manuscript with Track Changes.

L353: There is no need to start a new paragraph because here presented another reason for the non-significant correlation between WUE and plant NPK.

Response:

Thank you for your suggestion! We have merged the two paragraphs. Please see line 368 at the Revised Manuscript with Track Changes.

L364-370: These three sentences can be simplified into one sentence. For example: “The correlations between plant NPK varied with water addition levels (Fig. 2), N addition levels (Fig. 3), as well as the interaction between N addition and watering (Fig. 4). There results confirm …”

Response:

Thank you for your suggestion! We have revised the sentence. Please see lines 380-390 at the Revised Manuscript with Track Changes.

L375: change to “in that case”

Response:

Thank you for your suggestion! We have revised the sentence. Please see line 392 at the Revised Manuscript with Track Changes.

L380-381: change “on the other hand” to “in contrast”. Also is there any reference or explanation on this description that plant demands for P is greater than that for N when precipitation increases (the authors did not point out which direction of the change in precipitation)?

Response:

Thank you for your suggestion! We have revised the sentence. Please see line 397 at the Revised Manuscript with Track Changes. We made some mistakes in this description. What we want to express is that H. ammodendron in the present study may demand more P than N under precipitation increase. The reason is that the N/P ratio in H. ammodendron was mostly higher than 16, which have been mentioned later (lines 366-371). We have revised this description. Please see lines 403-404 at the Revised Manuscript with Track Changes.

L382-384: Bad grammar. Change to “… indicator of the relative limitation of N versus P in ecosystems, i.e., N/P ratio less than 14 generally indicates N limitation while N/P ratio greater than 16 suggests P limitation.”

Response:

Thank you for your suggestion! We have revised the sentence. Please see lines 399-402 at the Revised Manuscript with Track Changes.

L384-385: “almost higher than 16”? Do you mean “mostly higher than 16”? From Table S3, all mean N/P ratios were higher than 16. Also this should be mentioned in the Results section.

Response:

Thank you for your suggestion! We have change “almost” to “mostly”. Please see line 403 at the Revised Manuscript with Track Changes. Also, the result of N/P ratio mentioned here is to explain that the demand for P is higher than the demand for N in H. ammodendron in the present study, not the key results of the present study. So we did not present it in the Results section. 

L394: Where did the “interaction of temperature and precipitation” come from?

Response:

Sorry for our mistake. This sentence should be “interaction of N deposition and precipitation”. We have revised it. Please see line 414 at the Revised Manuscript with Track Changes.

Reviewer #2: This is the second time I have reviewed this manuscript, and I feel the authors have really improved the manuscript and limited the scope of their results to accurately reflect what the data are indicating. but, the standard of English is poor - even in this revision. Providing this is done, the manuscript is in my opinion acceptable for publication.

Response:

Thank you for your suggestion. We have asked an English editing company (Editage) to edit English.

---

## [Decision Letter · Decision Letter 2]

21 Jul 2021

PONE-D-21-06220R2

Changes in precipitation and atmospheric N deposition affect the correlation between N, P and K but not the coupling of water-element in Haloxylon ammodendron

PLOS ONE

Dear Dr. Wang,

Thank you for submitting your manuscript to PLOS ONE. After careful consideration, we feel that it has merit but does not fully meet PLOS ONE’s publication criteria as it currently stands. Therefore, we invite you to submit a revised version of the manuscript that addresses the points raised during the review process.

Please pay attention to the citation of equations in Discussion. You have deleted them in Materials and methods.

The format of Reference need to be checked carefully.

We look forward to receiving your revised manuscript.

Kind regards,

Yajuan Zhu, Ph.D.

Academic Editor

PLOS ONE

Journal Requirements:

Additional Editor Comments (if provided):

The author had deleted Measurements of gas exchanges and calculations of water use efficiency in Materials and method. However, in L282, there are citations of two equations in Discussion: "According to the definitions of ins-WUE and int-WUE (equations (1) and (2))," Please modify discussion according to Materials and method.

The full comma "." was missing in Line 332.

There are some format mistakes in References. In 21, 22, 25, 37, 39, 50, 54 and 55, all Latin name in title should be italic. In 22, genus name should be Artemisia and Caragana. In 50, only the first letter of the first word need to be capital. Please check them carefully.

Reviewers' comments:

Reviewer's Responses to Questions

**Comments to the Author**

1. If the authors have adequately addressed your comments raised in a previous round of review and you feel that this manuscript is now acceptable for publication, you may indicate that here to bypass the “Comments to the Author” section, enter your conflict of interest statement in the “Confidential to Editor” section, and submit your "Accept" recommendation.

Reviewer #1: All comments have been addressed

Reviewer #2: All comments have been addressed

2. Is the manuscript technically sound, and do the data support the conclusions?

Reviewer #1: Yes

Reviewer #2: Yes

3. Has the statistical analysis been performed appropriately and rigorously? 

Reviewer #1: Yes

Reviewer #2: Yes

4. Have the authors made all data underlying the findings in their manuscript fully available?

Reviewer #1: Yes

Reviewer #2: Yes

5. Is the manuscript presented in an intelligible fashion and written in standard English?

Reviewer #1: Yes

Reviewer #2: Yes

6. Review Comments to the Author

Reviewer #1: The manuscript was carefully revised based on the comments raised by both reviewers, and the writing has been greatly improved. I feel that this ms is now suitable for publication in PLOS One.

Reviewer #2: (No Response)

7. PLOS authors have the option to publish the peer review history of their article (what does this mean?). If published, this will include your full peer review and any attached files.

Reviewer #1: No

Reviewer #2: No

---

## [Author Response · Author response to Decision Letter 2]

6 Aug 2021

Journal Requirements:

Responses:

Thank you for your suggestion. We have reviewed our reference list. We have not cited any retracted papers. In the last revision, we have added 1 reference, it is reference [46]. However, we did forget to mention this change in the Responses to the Reviewers of the previous version. Sorry for our mistake in the last revision.

Additional Editor Comments (if provided):

The author had deleted Measurements of gas exchanges and calculations of water use efficiency in Materials and method. However, in L282, there are citations of two equations in Discussion: "According to the definitions of ins-WUE and int-WUE (equations (1) and (2))," Please modify discussion according to Materials and method.

Responses:

Thank you for your suggestion. We have revised the sentence. Please see lines 281-285 at the Revised Manuscript with Track Changes.

The full comma "." was missing in Line 332.

Responses:

Thank you for your suggestion. We have added the full comma “.”. Please see line 334 at the Revised Manuscript with Track Changes.

There are some format mistakes in References. In 21, 22, 25, 37, 39, 50, 54 and 55, all Latin name in title should be italic. In 22, genus name should be Artemisia and Caragana. In 50, only the first letter of the first word need to be capital. Please check them carefully.

Responses:

Thank you for your suggestion. We have checked the References and revised the mistake. Please see the References section at the Revised Manuscript with Track Changes.

---

## [Editor Report · Decision Letter 3]

10 Sep 2021

PONE-D-21-06220R3Changes in precipitation and atmospheric N deposition affect the correlation between N, P and K but not the coupling of water-element in Haloxylon ammodendronPLOS ONE

Dear Dr. Wang,

Thank you for submitting your manuscript to PLOS ONE. After careful consideration, we feel that it has merit but does not fully meet PLOS ONE’s publication criteria as it currently stands. Therefore, we invite you to submit a revised version of the manuscript that addresses the points raised during the review process.

Please pay attention to the citation of equations in Discussion. You have deleted them in Materials and methods.

The format of Reference need to be checked carefully.

We look forward to receiving your revised manuscript.

Kind regards,

Yajuan Zhu, Ph.D.

Academic Editor

PLOS ONE

Journal Requirements:

Additional Editor Comments (if provided):

Please pay attention to the citation of equations in Discussion. You have deleted them in Materials and methods.

The format of Reference need to be checked carefully.
---

## [Author Response · Author response to Decision Letter 3]

7 Oct 2021

Responses:

Thank you for your suggestion. We have reviewed our reference list. We have not cited any retracted papers. In the last revision, we have added 1 reference, it is reference [46]. However, we did forget to mention this change in the Responses to the Reviewers of the previous version. Sorry for our mistake in the last revision.

Additional Editor Comments (if provided):

The author had deleted Measurements of gas exchanges and calculations of water use efficiency in Materials and method. However, in L282, there are citations of two equations in Discussion: "According to the definitions of ins-WUE and int-WUE (equations (1) and (2))," Please modify discussion according to Materials and method.

Responses:

Thank you for your suggestion. We have revised the sentence. Please see lines 281-285 at the Revised Manuscript with Track Changes.

The full comma "." was missing in Line 332.

Responses:

Thank you for your suggestion. We have added the full comma “.”. Please see line 334 at the Revised Manuscript with Track Changes.

There are some format mistakes in References. In 21, 22, 25, 37, 39, 50, 54 and 55, all Latin name in title should be italic. In 22, genus name should be Artemisia and Caragana. In 50, only the first letter of the first word need to be capital. Please check them carefully.

Responses:

Thank you for your suggestion. We have checked the References and revised the mistake. Please see the References section at the Revised Manuscript with Track Changes.

---

## [Editor Report · Decision Letter 4]

11 Oct 2021

Changes in precipitation and atmospheric N deposition affect the correlation between N, P and K but not the coupling of water-element in Haloxylon ammodendron

PONE-D-21-06220R4

Dear Dr. Wang,

We’re pleased to inform you that your manuscript has been judged scientifically suitable for publication and will be formally accepted for publication once it meets all outstanding technical requirements.

Kind regards,

Yajuan Zhu, Ph.D.

Academic Editor

PLOS ONE

Additional Editor Comments (optional):

The authors have modified all questions and format mistakes. Now it's ready for publish.
---

## [Editor Report · Acceptance letter]

15 Oct 2021

PONE-D-21-06220R4 

Changes in precipitation and atmospheric N deposition affect the correlation between N, P and K but not the coupling of water-element in *Haloxylon ammodendron*

Dear Dr. Wang:

I'm pleased to inform you that your manuscript has been deemed suitable for publication in PLOS ONE. Congratulations! Your manuscript is now with our production department. 

Kind regards, 

on behalf of

Dr. Yajuan Zhu 

Academic Editor

PLOS ONE